# Study on the Effects of Biologically Active Amino Acids on the Micellization of Anionic Surfactant Sodium Dodecyl Sulfate (SDS) at Different Temperatures

Fatima M. Elarbi [1], Zaineb O. Ettarhouni [1] , Wanisa Abdussalam-Mohammed [2] and Aysha B. Mezoughi [1],*

[1]  Department of Chemistry, Faculty of Science, University of Tripoli, Tripoli P.O. Box 13503, Libya;
   rehanalrbei@gmail.com (F.M.E.); z.eltarhouni@uot.edu.ly (Z.O.E.)
[2]  Department of Chemistry, Faculty of Science, University of Sebha, Sebha P.O. Box 18758, Libya;
   Wan.Ahweelat@sebhau.edu.ly
*   Correspondence: abamezoughi2013@gmail.com

**Abstract:** The micellar properties of the anionic surfactant, sodium dodecyl sulfate (SDS) are modified by the biologically active amino acids. Amino acids (AAs) have experienced a variety of interactions and are proposed to influence SDS micelles due to their nominated hydrophobic interactions. The present study determines the critical micellar concentration (CMC) of SDS in aqueous solutions as well as in amino aqueous solutions. Three amino acids (glutamic acid, histidine, and tryptophan) are considered here. The conductometric measurements were carried out using a wide range of SDS concentrations at different temperatures. Surface tension experiments have also been applied to estimate many surface parameters including surface excess concentration ($\Gamma_{max}$), surface occupied area per surfactant molecule ($A_{min}$), surface tension at CMC ($\gamma_{cmc}$), surface pressure at CMC ($\Pi_{cmc}$) and Gibbs free energy of adsorption ($\Delta G^{\circ}_{ads}$), enthalpy $\Delta H^{\circ}_{m}$ and the critical packing parameter (*CPP*). Interestingly, CMC values of SDS in water and in aqueous amino acids estimated by the surface tension method are comparable with the corresponding values obtained by the conductance method. The thermodynamic parameters of SDS micellization were also evaluated in both presence and absence of AAs. The additives of AAs work to reduce the CMC values, as well as the SDS thermodynamic parameters. This reduction is highly dependent on the hydrophobicity of the AA side chain. Negative values of $\Delta G^{\circ}_{m}$, $\Delta H^{\circ}_{m}$ elucidate that micellization of SDS in the presence of amino acids is thermodynamically spontaneous and exothermic. The outcomes here might be utilized for pharmaceutical applications to stabilize proteins and inhibit protein aggregation.

**Keywords:** anionic surfactant; amino acids; sidechains; conductivity; surface tension; micellization; protein

## 1. Introduction

Surfactants are common molecules that exist in a wide range of commercial products, including detergents, cosmetics, herbicides, and some antibiotics [1,2]. They are also used to dissolve insoluble drugs as well as in drug delivery [3,4]. Moreover, surfactants are widely used in the food industry as emulsifiers to improve the absorption of lipid ingredients. Surfactant-coated nanoparticles play vital roles in food nanotechnology [5,6] These amphiphilic molecules are classified into nonionic and ionic surfactants (including; anionic, cationic, and amphoteric or zwitterionic). The most important class are the anionic surfactants, which are also called detergents, where their characteristic properties include high foaming and dispersing ability as well as protein denaturation. Generally, surfactants undergo self-assembly processes in aqueous solutions at a specific concentration called critical micelle concentration (CMC) forming micelles [7,8].

The surfactant micellization process is an interesting phenomenon that occurs due to hydrophobic and electrostatic interactions. Such aggregation is important for various phar-

maceutical, chemical, and biotechnological purposes [9,10]. The self-assembly of surfactant molecules is influenced by different parameters, such as pH, temperature, pressure, ion strength, surfactant structure, and additives [11,12].

Amino acids (AAs) are one of the most common additives as they possess peripheral charges, which make them strong structure breakers in aqueous solutions [13]. These biomolecules represent the basic structural units of proteins and are found in natural systems as well as in a wide spectrum of applications [14]. Having the common polar carboxylic and amino groups ($-COO^-$ and $NH_3^+$) leads to electrostatic interactions with charged species in an aqueous medium, such as ionic surfactants [15]. In addition, amino acids have different sidechains, which vary in size, charges, and hydrophobic and hydrophilic moieties. However, surfactant micellization could be affected by surfactant-amino acid interactions [16].

Some surfactants are considered as important additives to stabilize proteins while others cause protein denaturation. Therefore, it is essential to understand the protein-surfactant interactions in the aqueous medium. However, the direct thermodynamic study is infeasible due to the complexities of protein size and shape. Thus, investigation of the interactions between surfactants and the main protein units, amino acids, is crucial and provides a vital insight into conformational stability and protein folding behavior [7,17–20]. SDS is the most biologically important surfactant used in protein denaturation and purification of membrane lipids and membrane proteins. According to that, studying the interactions of this surfactant with essential amino acids can provide a diverse utilization of SDS for several biological purposes. Therefore, our aim in this work is to investigate the effect of three essential amino acids, glutamic acid (**Glu**), histidine (**His**), and tryptophan (**Trp**) on the SDS micellization process. The choice of the AAs presented here was based on the differences in their sidechain structures of (nonpolar **Trp**, polar acidic **Glu**, and polar basic **His**) (Figure 1) and their influences on the SDS micelles. Studies on the micellar behavior of SDS in aqueous amino acids have been reported, [1,3,21] nevertheless, the interactions of SDS with these particular AAs have not been previously evaluated.

**Glu**          **Trp**          **His**

**SDS**

**Figure 1.** Structure of amino acids at pH 7, glutamic acid (**Glu**), tryptophan (**Trp**), and histidine (**His**) and sodium dodecyl sulfate (SDS).

## 2. Materials and Methods

All chemicals were purchased from common commercial suppliers (Sigma-Aldrich, Gillingham, UK). SDS was purified by recrystallization from ethanol and then washed with diethyl ether and dried. AAs were used without further purification. All solutions were prepared in distilled deionized water. Stock solutions of 0.01 M of each AAs (L-Tryptophan, L-Histidine, and DL-Glutamic acid) were prepared in distilled water (pH 7.0) as solvent systems

to prepare SDS solutions. Due to the limited solubility of tryptophan in water, the chosen concentration of AAs is small. It is also a constant concentration during the experiment in order to make a fair comparison of the effect of AAs on the SDS micellization process. A stock solution of SDS was prepared and used to make a series of concentrations ranging from 1 to 16 mM for each of the AAs. The range of SDS concentrations covers the premicellar region, post micellar region, and critical micelle concentration. The conductivities of SDS in aqueous solutions of the three amino acids were measured using a conductivity/TDS meter (Jenway 4510, Chelmsford, UK) with a cell constant of 0.7475 cm$^{-1}$ at different temperatures (20, 25, 30, 35, 40, and 45 °C). The uncertainty in the conductance measurements was within ±0.5%. Conductometric measurements were carried out in a water thermostat bath WBT 12 (Wedingen, Schoemperlenstraße, 76185 Karlsruhe, Germany) with an accuracy of ±0.2 K. The conductivity data were plotted versus molar concentration of SDS both in the presence and absence of AAs and the break in the plot was observed. The concentration values that corresponded to the breakpoints were identified as the CMC values of SDS.

The surface tension measurements were carried out using the Du Noüy tensiometer model (Krüss K6, Darmstadt, Germany). Each sample solution was measured three times and the average value was used as surface tension.

## 3. Results

### 3.1. Conductometric Measurements

The interactions between SDS and amino acids have been studied at different temperatures. Conductometric measurements have been carried out over the range of SDS concentrations in aqueous solutions of 0.01 M of AAs (L-Tryptophan, L-Histidine, and DL-Glutamic acid) at 20, 25, 30, 35, 40, and 45 °C. Figure 3 shows the plots of electrical conductivity as a function of SDS concentration in the studied systems. The CMC values are obtained from the intersection point of the two straight lines of electrical conductivity versus the SDS concentration plot. Below the CMC there are no micelles and SDS molecules behave as a strong electrolyte and are fully dissociated in the aqueous solution. On the other hand, above the CMC monomer concentrations remain constant in a close value to the CMC, and the increase in the SDS concentration results in an increase in micelle concentration. The degree of micelle ionization ($\alpha$) is estimated from the ratio of the two slopes above (post- micellar region) and below CMC (pre- micellar region). Since the micelles behave as a weak electrolyte and are partially ionized, the variation of electrical conductivity after CMC is less compared to the values prior to the CMC [7,22,23]

The CMC values of SDS in water and in the presence of 0.01 M AAs at different temperatures are given in Table 1. The CMC of all studied systems increases with the increase in temperature. The reason is due to the increase in the dehydration of the charged hydrophilic groups of SDS molecules which causes electrostatic repulsions between the charged heads followed by an increase in the CMC [24]. The variation of CMCs with temperature is shown in Figure 2.

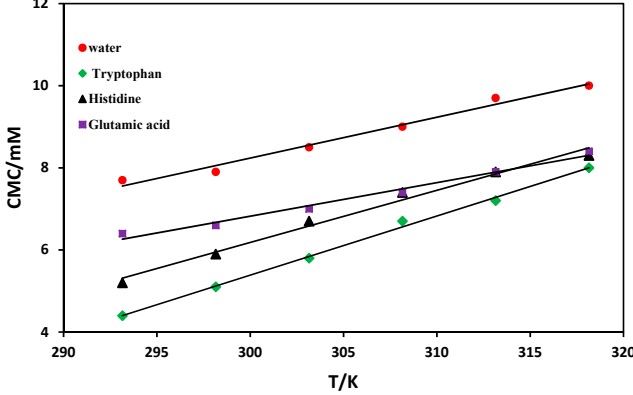

**Figure 2.** The CMC values of SDS in the presence and absence of amino acids as a function of temperature.

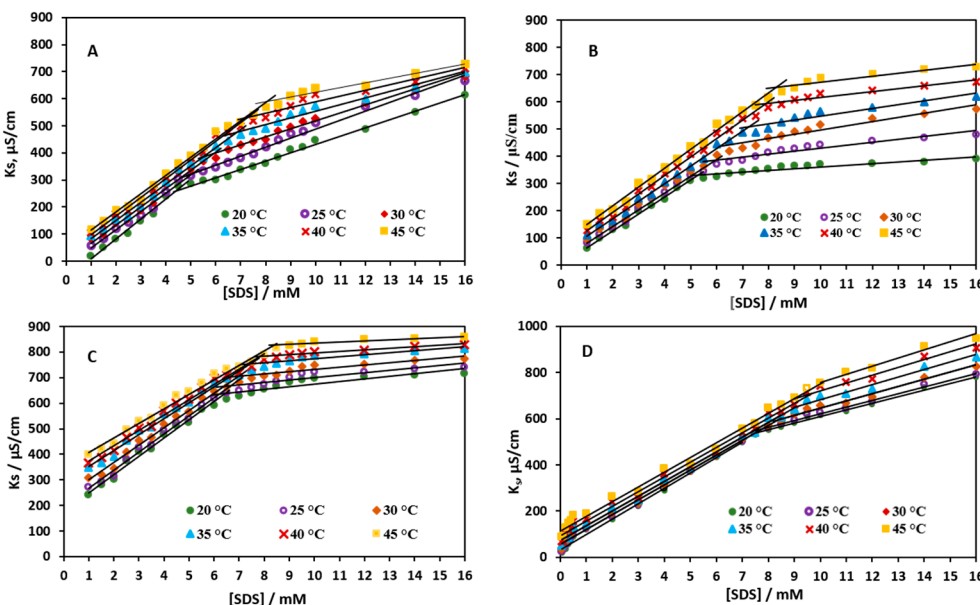

**Figure 3.** (**A**) The variation of electrical conductivity of SDS in an aqueous solution of 0.01 M L-Tryptophan with the concentration at different temperatures. (**B**) The variation of electrical conductivity of SDS in an aqueous solution of 0.01 M L-Histidine with the concentration at different temperatures. (**C**) The variation of electrical conductivity of SDS in an aqueous solution of 0.01 M DL-Glutamic acid with the concentration at different temperatures. (**D**) The variation of electrical conductivity of SDS with the concentration in water at different temperatures [25].

Amino acids are considered as water structure breaker solutes. Their influences on the CMCs of surfactant aqueous solutions can be explained in two ways considering the nature of these additives. As a result of the formation of hydrogen bonds between AAs and water molecules, a higher portion of AAs remain in the surrounding polar head group of the SDS micelle indicating significant interactions between SDS and amino acid molecules. On the other hand, the presence of amino acid molecules in the aqueous solution of SDS causes a break for the structured water molecules around the hydrophobic chains of SDS monomers. This results in an increase in the hydrophobicity of SDS molecules and consequently the entropy of the studied systems and makes the micellization of SDS occur at lower concentrations compared to the corresponding CMCs in the water [21,26].

**Table 1.** Values of critical micelle concentrations, CMCs of SDS (mM) in water and 0.01 M aqueous solutions of Tryptophan, Histidine, and Glutamic acid at different temperatures obtained from conductometric measurements.

| T/°C | 20 | 25 | 30 | 35 | 40 | 45 |
|---|---|---|---|---|---|---|
| Water [27] | 7.70 | 7.90 | 8.50 | 9.00 | 9.70 | 10.0 |
| DL-Glu | 6.40 | 6.60 | 7.00 | 7.40 | 7.90 | 8.40 |
| L-His | 5.20 | 5.90 | 6.70 | 7.40 | 7.90 | 8.30 |
| L-Trp | 4.40 | 5.10 | 5.80 | 6.70 | 7.20 | 8.00 |

CMC values of SDS at 0.01 M amino acid follow the order: Glu > His > Trp. This trend can be explained in terms of the hydrophobicity nature of these amino acids. The hydrophobicity of amino acids is proposed by Kyte and Doolittle and according to the obtained hydrophobicity index of the amino acids, tryptophan ($-0.9$) is more hydrophobic than that of histidine ($-3.2$) and glutamic acid ($-3.5$) [27].

The incorporation of amino acid molecules into micelles rises when the hydrophobic and the hydrophilic interactions increase and decrease, respectively. Consequently, the reduction of CMC is the highest in the presence of tryptophan among the three studied

amino acids. The reason is due to the high hydrophobicity of the tryptophan side chain compared to the other two amino acids. The decrease in CMC values in the presence of amino acids could also be interpreted by the suggested expansion in SDS micelles when the AAs sidechain incorporates with SDS tails inside the micellar core. This is noticeable in the tryptophan case due to its higher hydrophobicity, therefore, higher incorporation in the hydrophobic environment inside the micellar core. Studying the micellar shape and size is extremely required at this stage.

### 3.2. Surface Tension Measurements

The plots of surface tension versus SDS concentration at 20 °C are displayed in Figure 4. It has been shown that the surface tension of the studied systems decreases with increasing surfactant concentration up to a specific concentration, above which the surface tension remains constant. A breakpoint in the plots of surface tension versus concentration indicates the CMC value of SDS. The decrease in the surface tension with the increasing concentration is due to the hydrophobic effect which forces surfactant molecules to adsorb at the air/aqueous solution interface. At CMC the surface is occupied by the surfactant molecule, above the CMC any additional surfactant molecules are spontaneously associated to form micelles [28].

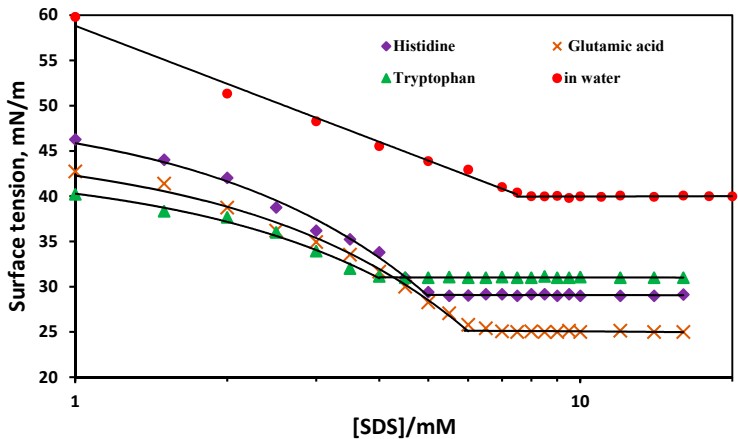

**Figure 4.** Surface tension plots of SDS solutions as a function of concentration in water and the 0.01 M of different AAs at 20 °C.

The estimated CMC values of different systems from surface tension measurements are reported in Table 2. Surface tension data have been used to calculate the surface properties of SDS aqueous solutions in the presence of 0.01 M amino acids, such as maximum surface excess concentration ($\Gamma_{max}$), surface occupied area per surfactant molecule ($A_{min}$), surface tension at CMC ($\gamma_{cmc}$), surface pressure at CMC ($\Pi_{cmc}$) and Gibbs free energy of adsorption ($\Delta G_{ads}^{\circ}$). The critical packing parameter (*CPP*) is also calculated to determine the shape of the micelles in the studied systems.

**Table 2.** Values of the CMC, maximum surface excess concentration, minimum occupied area per surfactant molecule, surface tension at CMC, surface pressure at CMC, Gibbs free energy of adsorption and, packing parameter of SDS in the presence and absence of 0.01 M amino acid at 20 °C.

| System | CMC/mM | $\Gamma_{max} \times 10^6$ mol/m$^2$ | $A_{min}$/Molecule nm$^2$ | $\gamma_{cmc}$ mN/m | $\Pi_{cmc}$ mN/m | $\Delta G_{ads}^{\circ}$ KJ/mol | *CPP* |
|---|---|---|---|---|---|---|---|
| Water [25] | 7.50 | 1.952 | 0.8507 | 40.20 | 32.00 | −50.34 | 0.247 |
| L-Glu | 6.20 | 2.023 | 0.8208 | 25.35 | 47.05 | −64.33 | 0.256 |
| L-His | 5.00 | 2.186 | 0.7596 | 29.40 | 43.40 | −62.79 | 0.276 |
| L-Trp | 4.00 | 2.028 | 0.8188 | 31.13 | 41.47 | −56.86 | 0.256 |

The maximum surface excess concentration, $\Gamma_{max}$ is a measure of the effectiveness of adsorption of the surfactant systems at air/aqueous solution interface and it can be calculated using Gibbs adsorption Equation (1) [28]:

$$\Gamma_{max} = -\frac{1}{nRT}\left(\frac{\partial\gamma}{\partial ln\ C}\right) \tag{1}$$

where; $\Gamma_{max}$ is the maximum surface excess concentration of the surfactant system, $n$ is the number of species produced in a solution by surfactant molecule (for SDS $n = 2$). $(\partial\gamma/\partial lnC)$ is the slope of the $\gamma$ against $ln[C]$ plot, $R$ is the gas constant, and $T$ is the temperature in absolute scale. The surface occupied area by each surfactant molecule at the interface can be calculated from the following Equation (2):

$$A_{min} = \frac{10^{18}}{N_A\ \Gamma_{max}} \tag{2}$$

where; $N_A$ is Avogadro's number.

As shown in Table 2 the value of $\Gamma_{max}$ of SDS in aqueous solutions of amino acids is higher than that in water, because of the electrostatic repulsions between charged head groups which are decreased in the presence of amino acids. As a result of amino acid hydrophobicity, some of the water molecules in the hydration layer around SDS molecules which are adsorbed at the interface are replaced by amino acids molecules. Consequently, electrostatic attractions between $-NH_3^+$ groups in the amino acids and negatively charged head groups of SDS molecules have occurred. Therefore, the SDS molecules are tightly packed at the interface, which increases the surface concentration of SDS and decreases the surface occupied area by SDS molecules compared to those in the absence of amino acids [1,21,29]. The effectiveness of surfactant systems to lower the surface tension is measured by calculating the surface pressure at CMC, $\Pi_{cmc}$ which is obtained from Equation (3) [28]:

$$\pi_{CMC} = \gamma_{water} - \gamma_{CMC} \tag{3}$$

where $\gamma_{water}$ is the surface tension of water and $\gamma_{CMC}$ is the surface tension of the surfactant solution at CMC. The values of $\Pi_{cmc}$ for the different amino acids systems are higher than the corresponding value in the absence of amino acids, indicating that the studied surfactant systems are more effective for lowering surface tension in the presence of amino acids.

Gibbs free energy of adsorption at the air/water interface, $\Delta G_{ads}^{\circ}$ is also evaluated using Equation (4):

$$\Delta G_{ads}^{\circ} = \Delta G_m^{\circ} - \frac{\pi_{CMC}}{\Gamma_{max}} \tag{4}$$

where $\Delta G_m^{\circ}$ is the corresponding Gibbs free energy of micellization of the system.

The Gibbs free energy of adsorption $\Delta G_{ads}^{\circ}$ at 20 °C of all studied systems is calculated, and the obtained values (Table 2) are negative which indicates that the adsorption of surfactant molecules at air/aqueous solution interface is a spontaneous process in the presence and absence of amino acids. As a result of the hydrophobic effect, some surfactant molecules are transferred to the interface and adsorb on it in order to decrease the free energy of the system, where the hydrophilic heads are directed to the aqueous solution and the hydrophobic tails are directed towards the air [28,30]. It is observed that $\Delta G_{ads}^{\circ}$ values are more negative in the presence of amino acids, which suggests that the adsorption process is more spontaneous than that in water [18]. Calculated $\Delta G_{ads}^{\circ}$ values are more negative than corresponding $\Delta G_m^{\circ}$, indicating that the adsorption process of surfactant molecules is more spontaneous and proceeds the micellization. Packing parameters of surfactant molecules in the micelles' *CPP* have been calculated from the following Equation (5) [31]:

$$CPP = \frac{V}{l_c \, A_{min}} \tag{5}$$

where $V$ is the volume of the hydrophobic tail, and $l_c$ is the length of the hydrophobic tail. $V$ and $l_c$ are given by Tanford's Equation (6) [31]:

$$V = 0.0274 + 0.0269n \tag{6}$$

$$l_c = 0.1500 + 0.1265n \tag{7}$$

where $n$ is the number of carbon atoms of the hydrophobic tail of the surfactant molecule. $V$ is in cubic nanometer (nm$^3$) and $l_c$ is in (nm). All obtained values of $CPP$ (Table 2) are in the range of spherical structures ($CPP < 1/3$), and as a result of the smaller surface occupied area of SDS molecules in the presence of amino acids, $CPP$ values in aqueous amino acid solutions are higher than corresponding $CPP$ in the absence of amino acids.

### 3.3. Thermodynamics of Micellization

The temperature dependence of the CMC is used to calculate the thermodynamic parameters of micellization.

Gibbs free energy ($\Delta G_m^{\circ}$), enthalpy ($\Delta H_m^{\circ}$), and entropy ($\Delta S_m^{\circ}$) of SDS micellization in the presence and absence of amino acids have been calculated from the following Equation (8) [28]:

$$\Delta G_m^{\circ} = (2 - \alpha)RT \, lnX_{cmc} \tag{8}$$

where; $X_{cmc}$ is the CMC of the system in the mol fraction unit, $\alpha$ is the degree of counterion dissociation of micelles, $R$ is the gas constant and the $T$ is the absolute temperature.

$$\Delta H_m^{\circ} = -(2 - \alpha)RT^2 \left[ \frac{\partial(lnX_{cmc})}{\partial T} \right] \tag{9}$$

The value of $\left[ \frac{\partial(lnX_{cmc})}{\partial T} \right]$ is obtained from the slope of $lnX_{cmc}$ against the $T$ plot.

$$\Delta S_m^{\circ} = \frac{\Delta H_m^{\circ} - \Delta G_m^{\circ}}{T} \tag{10}$$

The estimated values of $\alpha$, $\Delta G_m^{\circ}$, $\Delta H_m^{\circ}$, and $\Delta S_m^{\circ}$ at different temperatures are summarized in Table 3. The values of $\Delta G_m^{\circ}$ are negative for all studied systems which suggests that the micellization is a spontaneous process over the studied temperature range. More negative values of $\Delta G_m^{\circ}$ are observed in amino acid aqueous solutions, indicating that the micellization is more spontaneous in the presence of amino acids.

From $\Delta H_m^{\circ}$ values, it can be seen that the micellization of studied systems is an exothermic process and the value of $\Delta H_m^{\circ}$ of most systems becomes more exothermic as the temperature increases. The entropy of micellization, $\Delta S_m^{\circ}$ is positive over the investigated temperature range, and its value decreases with an increase in temperature. The positive $\Delta S_m^{\circ}$ values are associated with the destruction of the structured water molecules around the hydrophobic tails of SDS molecules when they transfer from the aqueous solution to the interior of the micelles. The values of $\Delta S_m^{\circ}$ of studied systems decrease with the increase in temperature as a result of reduction of the degree of hydration of hydrophobic tails of SDS with temperature [7,21,24,32].

**Table 3.** Thermodynamic parameters of SDS micellization in the absence and presence of 0.01 M amino acid at different temperatures.

| T/K | α | $\Delta G^{\circ}_m$ KJ/mol | $\Delta H^{\circ}_m$ KJ/mol | $\Delta S^{\circ}_m$ KJ/mol |
|---|---|---|---|---|
| Water | | | | |
| 293.15 | 0.428 | −34.04 | −12.69 | 0.07283 |
| 298.15 | 0.438 | −34.30 | −13.04 | 0.07131 |
| 303.15 | 0.487 | −33.50 | −13.06 | 0.06743 |
| 308.15 | 0.515 | −33.21 | −13.25 | 0.06477 |
| 313.15 | 0.538 | −32.94 | −13.47 | 0.06217 |
| 318.15 | 0.559 | −32.87 | −13.70 | 0.06025 |
| L-Trp | | | | |
| 293.15 | 0.418 | −36.41 | −26.90 | 0.03241 |
| 298.15 | 0.494 | −34.70 | −26.49 | 0.02754 |
| 303.15 | 0.438 | −34.83 | −28.40 | 0.02035 |
| 308.15 | 0.365 | −37.80 | −30.72 | 0.02297 |
| 313.15 | 0.307 | −39.46 | −32.85 | 0.02111 |
| 318.15 | 0.256 | −40.81 | −34.93 | 0.01848 |
| L-His | | | | |
| 293.15 | 0.101 | −42.94 | −25.64 | 0.05901 |
| 298.15 | 0.176 | -41.37 | −25.48 | 0.05330 |
| 303.15 | 0.253 | −39.73 | −25.23 | 0.04783 |
| 308.15 | 0.225 | −40.58 | −26.48 | 0.04576 |
| 313.15 | 0.161 | −42.41 | −28.34 | 0.04493 |
| 318.15 | 0.163 | −42.80 | −29.22 | 0.04268 |
| L-Glu | | | | |
| 293.15 | 0.142 | −41.07 | −14.87 | 0.08937 |
| 298.15 | 0.133 | −41.83 | −15.45 | 0.08848 |
| 303.15 | 0.132 | −42.28 | −15.99 | 0.08672 |
| 308.15 | 0.128 | −42.80 | −16.55 | 0.08519 |
| 313.15 | 0.102 | −43.78 | −17.33 | 0.08446 |
| 318.15 | 0.078 | −44.74 | −18.12 | 0.08367 |

## 4. Conclusions

In conclusion, the effect of **Glu, His**, and **Trp** on the micellization of SDS has been investigated at various temperatures. It was observed that the behavior of these additives depends on their nature at a fixed temperature. The CMC values in the presence of additives were lower compared to in pure water. This reduction may be attributed to the hydrogen bonding between AAs and water molecules when these additives are presented in the outer portion of the SDS micelle. The decrease of CMC in the case of Trp is more than that of His and Glu. The possible reason is probably due to the higher hydrophobicity of the Trp sidechain, which is a nonpolar amino acid, whereas His and Glu are basic and acidic polar amino acids. Therefore, Trp can promote the formation of SDS micelles more efficiently than His and Glu. SDS micellar core expansion in the presence of AAs was suggested to be another element to elucidate this reduction. Further investigations including small angle neutron scattering and fluorescence are necessitated to accomplish this discussion.

**Author Contributions:** Conceptualization, A.B.M. and Z.O.E.; methodology, F.M.E.; Writing an original draft, A.B.M., Z.O.E. and F.M.E.; writing-review and editing Z.O.E., W.A.-M. and A.B.M. All authors have read and agreed to the published version of the manuscript.

**Funding:** This research received no external funding.

**Data Availability Statement:** Data are contained within the article.

**Acknowledgments:** We would like to express our sincere appreciation to the University of Tripoli and University of Sebha for provided resources and services.

**Conflicts of Interest:** Authors have declared that no conflict of interest exist.

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
