# Peer review of "Study on the Effects of Biologically Active Amino Acids on the Micellization of Anionic Surfactant Sodium Dodecyl Sulfate (SDS) at Different Temperatures"

_chemistry, doi:10.3390/chemistry4010013_

Round 1

Reviewer 1 Report

The behavior of amino acids and surfactants has attracted interest from a wide range of scientific fields. In this study, the authors have tried the relationship between SDS micelle behavior with a particular focus on individual amino acids from a physicochemical point of view. The obtained results will provide valuable and useful information to researchers in this area. Experimental results are clear, and the story is carefully and logically described and well presented. The reviewer found the results obtained in this article are valuable, but the reviewer suggests the following comments to help the reader better understand the potential benefits of this article.

Comment 1

In this study, individual amino acids (glutamic acid, histidine, and tryptophan) are used as models for amino acids. The reviewer recommends adding one more figure that explains the chemical structure of these three amino acids and the surfactant (SDS). The addition of this new figure will help the reader to better visualize the phenomena revealed in this paper.

Comment 2

The authors said in lines 30 and 31 that "They are also used to dissolve insoluble drugs as well as in drug delivery [3,4].". Recently, surfactants have been remarkably used not only in these areas but also in the field of food chemistry. Reviewer recommend to add one sentence explaining that surfactants are also a key factor in the area of food chemistry. Such an explanation would make this paper more interesting for food chemistry researchers like reviewer (me). For example, an extensive review of food chemically active surfactants has been reported, including the following:

Front. Sustain. Food Syst. 2019, 3:95

Nanoemulsions and their potential applications in food industry

https://www.frontiersin.org/articles/10.3389/fsufs.2019.00095/full

doi: 10.3389/fsufs.2019.00095

Int J Nanomedicine. 2021, 16:3937-3999

A Critical Review of the Use of Surfactant-Coated Nanoparticles in Nanomedicine and Food Nanotechnology

https://www.dovepress.com/a-critical-review-of-the-use-of-surfactant-coated-nanoparticles-in-nan-peer-reviewed-fulltext-article-IJN

doi: 10.2147/IJN.S298606

Author Response

Comment 1

In this study, individual amino acids (glutamic acid, histidine, and tryptophan) are used as models for amino acids. The reviewer recommends adding onnoe more figure that explains the chemical structure of these three amino acids and the surfactant (SDS). The addition of this new figure will help the reader to better visualize the phenomena revealed in this paper.

Our response: We have added figure (Figure 1) in the revised manuscript included the chemical structure of three amino acids at pH 7 as well as SDS structure.

Comment 2

The authors said in lines 30 and 31 that "They are also used to dissolve insoluble drugs as well as in drug delivery [3,4].". Recently, surfactants have been remarkably used not only in these areas but also in the field of food chemistry. Reviewer recommend to add one sentence explaining that surfactants are also a key factor in the area of food chemistry. Such an explanation would make this paper more interesting for food chemistry researchers like reviewer (me). For example, an extensive review of food chemically active surfactants has been reported, including the following:

Our response: We have added the applications of surfactants in food industry on lines 31, 32, 33 of the revised manuscript and we have also added the mentioned papers to the references.

Reviewer 2 Report

In this manuscript, Mezoughi and coworkers are representing an interesting experimental finding with related to the effects of the biologically important amino acids towards the micellization of SDS. This topic can be quite interesting to the biophysics and structural biology community. The present form of the manuscript is not very well organized, and authors may need to improve their writing and the organization of the content to deliver more coherent and convincing output to the reader community.  Authors do not emphasize their hypothesis why they are conducting certain experiments and what will be their expected outcome vs experimental results. This is a critically important step of scientific writing. I believe the topic of the manuscript is interesting, but authors may require providing a major revision for further consideration this this journal. Specifically, I would recommend authors to pay close attention to following points:

(1). What is the specific application of this finding? can authors highlight this point in the abstract?

(2). The abstract of the manuscript is not well organized. Can authors carefully revised their abstract to highlight the importance of their findings?

(3). Why did authors specifically choose biologically active amino acids and what is the logic behind this selection?

(4). Can authors provide a sample figure/cartoon to demonstrate how these amino acids affect the micellization in solution?

(5). Authors are using 10 mM AA concentration and 1 to 16 mM SDS concentration during these experiments? Why specifically choose these concentrations? Can authors be more elaborative? 

(6). There are multiple typing errors while writing centigrade and there are some other grammatical and textual errors. I would strongly recommend authors to carefully read and revise their manuscript writing.

(7). I think authors should combined figures 1 and 2 (a) as they both refer to the same experiment. In addition, can authors include the data for water here as the figure 1(d)?

(8). The CMC values as a function of SDS concentration seems interesting. can authors explain why CMC values in water is high?  can authors calculate the slopes for each derivative in the figure 3?

(9). Using only 10 mM AA concentration for this experiment is not sufficient. can authors also use one lower and one higher concentration of the AA for this experiment and provide the data?

(10). Also, with respect to the figure 4; How does altering AA concentration affect here?

(11). Can authors explain how does CPP value affect in a real system?

(12). Can authors plot a graph of ∝ vs temperature and explain the outcome how affect the micellization?

(13). Specifically, can authors describe why this study is important and what is the application of these findings in biological environments?

Round 2

Reviewer 2 Report

Authors have reasonably answered my suggestions. Even though they haven't answered all my suggestions, I believe the current format of the manuscript is acceptable.

But I would also suggest a final grammar check prior to the publication.

Author Response

The manuscript has been revised